# Determinants of Exposure to Potentially Toxic Metals in Pregnant Women of the DSAN-12M Cohort in the Recôncavo Baiano, Brazil

**DOI:** 10.3390/ijerph20042949

**Published:** 2023-02-08

**Authors:** Homègnon A. Ferréol Bah, Victor O. Martinez, Nathália R. dos Santos, Erival A. Gomes Junior, Daisy O. Costa, Elis Macêdo Pires, João V. Araújo Santana, Filipe da Silva Cerqueira, José A. Menezes-Filho

**Affiliations:** 1Institute of Collective Health, Federal University of Bahia, Salvador 40110-040, Brazil; 2Laboratory of Toxicology, College of Pharmacy, Federal University of Bahia, Salvador 40170-115, Brazil; 3Graduate Program in Pharmacy, College of Pharmacy, Federal University of Bahia, Salvador 40170-115, Brazil; 4Graduate Program in Food Science, College of Pharmacy, Federal University of Bahia, Salvador 40170-115, Brazil

**Keywords:** environmental exposure, toxic metals, determinants, pregnancy, maternal health

## Abstract

Exposure to potentially toxic metals (PTM) threatens maternal and child health. We investigated the determinants of exposure to lead (Pb), cadmium (Cd), arsenic (As), and manganese (Mn) in 163 pregnant women from the Recôncavo Baiano, Brazil, enrolled in the DSAN-12M cohort. We measured these metals in biological samples (blood, toenails, and hair) and the Pb dust loading rates (RtPb) at their homes by graphite furnace atomic absorption spectrophotometry (GFAAS). Questionnaires were applied to collect sociodemographic and general habits data. Only 2.91% (*n* = 4) of the pregnant women had As levels above the detection limit. Few participants had levels above the recommended reference values for blood Pb (5.1; 95% CI: 2.1–10.1%), and Mn in hair or toenails (4.3; 95% CI: 2.3–10.1%). On the other hand, 61.1 (95% CI: 52.4–69.3%) had elevated blood Cd levels. After binary logistic regression, low socioeconomic status, domestic waste burning, being a passive smoker, multiparity, and renovating the house significantly increased the chances of having high levels of Mn, Pb, and Cd. We detected a worrying situation related to exposure to Cd, showing the urgency of implementing human biomonitoring in the general population, especially in situations of social vulnerability.

## 1. Introduction

Environmental pollutants such as potentially toxic metals (PTM) have been associated with deleterious effects on human health depending on the magnitude of exposure [1]. Exposure to a high level of PTM such as lead (Pb), arsenic (As), cadmium (Cd) and the micronutrient manganese (Mn) have been the subject of several investigations in workers from the industrial and mining sector and in populations living in risk zones [2,3,4,5,6]. Over time, however, the neurotoxic potential of these metals, even at low exposure levels in children have been reported, [7,8,9,10] and there is a gap of knowledge regarding the toxicity of such contaminants in women [2]. In pregnant women, there is also a growing concern due to their vulnerability and the possibility of transferring PMT to the fetus; thus, compromising the child’s development [10,11].

During the uterine phase and in childhood, the plasticity of the central nervous system (CNS), the immaturity of defense mechanisms, and fetal and child elimination mechanisms make them more susceptible than adults [12]. For example, low levels of exposure to Pb, As, and Cd have been associated with high blood pressure, changes in kidney function, and increased risk of preeclampsia for pregnant women, in addition to impairing the neurodevelopment of the fetus and child [13,14,15]. In the fetus, the immaturity of Mn elimination mechanisms explains its higher concentration when compared to the mother’s in the umbilical cord, which may impair its cognitive development [10].

Studies have demonstrated possible interactions between PTM as a possible way to explain their toxicity in case of concomitant exposure [11,16,17,18]. Social or sociodemographic determinants may also favor exposure to these contaminants in vulnerable populations [19] in addition to acting conjointly with them to cause more exacerbated deleterious effects on health in general and to the CNS, specifically [20]. Although these metals have different physiological profiles, they show neurotoxic potential for children. Pb, As, and Cd are xenobiotics with no physiological role in the body and are considered threats to the kidney, liver, nervous system, and other organs [11,21]. Mn is an essential micronutrient whose excess has been associated with neurological damage, especially in children [9,22], while the occurrence of deleterious effects at low exposure levels is subject to inconsistency in the literature [23]. Therefore, there is a constant need for environmental and biological monitoring to detect possible sources of human exposure and raise awareness among populations about determinants and risk factors. Environmental components (water, soil, air) and biological matrices (blood, urine, nails, and hair) may serve that purpose [5,19,24].

These metals were chosen because they are naturally part of the environment, the need for more information on exposure in a population not occupationally exposed in the Brazilian context, and the health risk even at low levels. Also, PTM levels found in pregnant women are considered a proxy for intrauterine exposure. Previously, we reported moderate to high levels of PTM (Pb, Mn, and Cd) in the population (adults and school-age children of both sexes) of socially vulnerable communities in the same region [3,4,5,16,19,24]. However, although the PTM represents a threat during pregnancy, little is known about their real impact in socially vulnerable populations.

Considering women and fetus vulnerability during pregnancy when exposed to environmental contaminants, we carried out the study “Socioenvironmental Determinants of Child Neurodevelopment” (DSAN-12M), which is a birth cohort in the municipalities of Aratuípe and Nazaré das Farinhas in the Recôncavo Baiano, Brazil. The aims of this work are (i) to evaluate the exposure to PTM of the recruited pregnant women of the DSAN cohort, and (ii) to investigate the factors associated with such exposures.

## 2. Materials and Methods

### 2.1. Population and Study Design

This study is part of the DSAN-12M study in Aratuípe and Nazaré das Farinhas in the Recôncavo Baiano. These municipalities are inserted in a context of vulnerability due to unfavorable social conditions and the coexistence of some families with risky craft activities such as the production of lead-glazed ceramics, agricultural activities, and proximity to palm oil and soap factories, and quarries. The DSAN-12M assesses the impact of prenatal and postnatal environmental exposure to environmental contaminants (PTM, pesticides, and pathogens), maternal mental health and social factors that may impact children’s neurodevelopment. In this cross-sectional study, we investigate the intensity of exposure to Pb, As, Cd, and Mn in a cohort of pregnant women, using environmental and biological matrices.

Field work started in July 2019 but stopped early March 2020 due to the COVID-19 pandemic. A second collection phase occurred between July 2021 thru September 2022.

Ethical issues: This project was approved by the research ethics committee of the Faculty of Pharmacy-UFBA through Resolution 466-CNS/2012, with approval No. 3246555.

#### 2.1.1. Recruitment

Between 2006 and 2016, these two municipalities had an annual live birth average of 555 [25]. To invite as many pregnant women as possible, the field collection was carried out with the network of primary care units (PCU) of the Brazilian Public Health System (SUS is its acronym in Portuguese) in the municipalities of Nazaré (11 units) and Aratuípe (4 units) that carry out prenatal consultations. Considering the acceptance rate (70%) in other surveys [19,26] in the same region, the expected number of participants was 390 pregnant women.

During the first prenatal consultation, the pregnant women in the first or second trimester were informed about the study aims and methods by the nurses and asked to meet with the field investigator staff. A detailed project explanation was presented during the visit to the participant’s home. After the acceptance of the pregnant woman, the informed consent term was signed. In the case of a participant under 18 years, the legal guardian was asked to sign while the minor provided a written agreement.

As inclusion criteria, we selected pregnant women with a gestational age of fewer than 24 weeks who started their prenatal consultation at the PCU of the municipalities; lived in the region for at least one year before the pregnancy. Women with twin pregnancies under the prescription of medication potentially of neurotoxic risk for the fetus were excluded from the study.

#### 2.1.2. Socioeconomic Data

Trained interviewers applied questionnaires to pregnant women. The socioeconomic level (SES), stratified into five categories from A to E, was defined based on the criteria of the Brazilian Association of Population Studies [27]. Data were collected on general daily habits, education level, diet, occupational history, work during pregnancy, parity, history of active or passive smoking, and proximity to probable sources of exposure to potentially toxic metals.

### 2.2. Assessment of Exposure to PTM

Pregnant women’s exposures were measured using environmental and biological samples collected in the second trimester (between weeks 12 and 24). Domestic settled dust was used as an environmental indicator to estimate Pb exposure, which represents an indoor source. We calculated the Pb dust loading rate, while the biological samples used were blood (for Cd and Pb), hair (for Mn and As), and toenails (for Mn).

#### 2.2.1. Assessment of Pb Dust Loading Rate (RtPb)

Details on the sampling method and Pb determination have been described elsewhere [19,28]. Briefly, in the participant’s home, three disposable samplers (polyethylene Petri dishes) were installed on a support at a height of two meters in different rooms and left open for at least 30 days. After this period, the samplers were recovered and sent to the laboratory. The dust collected was solubilized in 40 mL of 7M ultrapure nitric acid and transferred to a volumetric flask of 50 mL and completed with ultrapure water. Pb analysis was performed by graphite furnace atomic absorption spectrophotometry (GFAAS) on an AA240Z, GTA-120 equipment (Varian^®^, Palo Alto, CA, USA). Pb levels were expressed as RtPb (µg Pb/m^2^/30 days), following the methodology described in Menezes-Filho et al. [28].

#### 2.2.2. Assessment of Biomarkers

##### Biological Sample Collection

Blood samples were collected by cubital venipuncture in vacuum tubes. Trace Elements Sodium Heparin tubes (Vacuette^®^, São Paulo, Brasil) were used for Pb and Cd determination. The samples were packed into isothermal boxes containing recyclable ice and transported to the Laboratory of Toxicology (Labtox) at the Faculty of Pharmacy of the Federal University of Bahia.

Occipital hair and toenail samples were collected with stainless steel tools (scissors and nail clipper). The first two centimeters of the hair tuft were used for Mn and As determinations as described by Menezes-Filho et al. [3]. The toenail polish was removed with an acetone-based solution before collection.

##### Blood Lead Level (PbB)

Blood samples were analyzed by GFAAS as described by Menezes-Filho et al. [29]. The Pb concentration was determined from the calibration curve obtained by diluting a standard Pb solution at 1000 µg/mL diluted in HNO_3_ (2%). Samples in duplicates, calibrators and reference material for quality control were diluted (1 + 9). Samples from the Proficiency Program for Blood Lead Analysis at Instituto Adolfo Lutz were analyzed concomitantly to ensure the quality of the analytical method. The precision and accuracy obtained were 8.8% and 107.4%, respectively. The method’s limit of detection (LOD) was 0.1 µg/dL. Samples with a PbB concentration below LOD were entered into the database with an LOD/2 value (i.e., 0.05 µg/dL).

##### Blood Cadmium Level (CdB)

CdB determinations were performed according to Kummrow et al. [30]. Briefly, 100 μL of the whole blood sample and certified reference material (Bio-Rad Lyphocheck^®^, Irvine, CA, USA) Whole Blood Metal Control Level 1) were transferred into Eppendorf^®^ (Guangzhou, China) microtubes with 200 μL of 0.4% Triton X-100 solution and 100 µL of 3 M HNO_3_. After mixing by vortex and centrifugation in a Sigma^®^ (Osterode am Harz, Germany) microcentrifuge (at 15,183 g) for 15 min, the supernatant was analyzed by GFAAS. Quality control samples were reanalyzed every ten samples. The precision and accuracy obtained were 7.6% and 19.5%, respectively. The LOD was set at 0.1 μg/L, and results below this limit were entered into the dataset as LOD/2.

##### Hair and Toenail Mn (MnH and MnTn) and Hair As (AsH) Levels

Hair and toenail samples were cleansed with non-ionic detergent solution (Merck^®^ (Darmstadt, Germany; Triton X-100; 1%) in an ultrasonic bath following the procedure reported by Dos Santos et al. [31]. The samples were dried at 60 °C for 3 to 4 h and approximately 50 mg were digested with 3 mL of spectroscopic grade concentrated nitric acid (JT Baker^®,^ Deventer, The Netherlands) using the Mars-Express6 microwave (CEM, Dallas, Texas, USA). The digestion process was carried out according to the conditions specified by the manufacturer. Each digestion run was carried out with a reagent blank, and the certified reference materials under the same conditions as the samples to ensure the analytical quality of each run.

After complete digestion, the solution was transferred into graduated polypropylene centrifuge tubes (Corning^®^, St. Louis, MO, USA) and volume adjusted to 10 mL with ultrapure water.

*MnH and MnTn:* All samples were processed in duplicate while the reference material (IAEA-085 human hair) was analyzed in every twenty readings. Mn levels were determined by GFAAS. The results were expressed in µg of Mn/g of hair (MnH) or nails (MnTn). The LOD was set at 0.1 μg/L, and results from samples below this limit were included in the data set as LOD/2.

*AsH*: In the case of As, the process was the same as for Mn with the certified reference material, rice flour (SRM 1568b Rice Flour, NIST, Gaithersburg, MD, USA), used for quality assurance. The results were expressed in µg of As/g of hair. The LOD value was 0.09 μg/L, and results below this limit were included in the data set as LOD/2.

Analytical accuracy was estimated in the range of 100 and 104%. The intra- and inter-run precision was 1.5% and 3.3%, respectively.

### 2.3. Data Analysis

The absolute and relative frequencies of the main sociodemographic variables of the participants were presented. The distributions of continuous variables (such as age and PTM concentrations) were evaluated using the Kolmogorov–Smirnov (KS) or Shapiro–Wilk (SW) test, and they were described as mean (±standard deviation) and median (interquartile range).

The biomarker values were dichotomized for two purposes: first, to define the proportions (95%; confidence interval) of participants with PTM levels above the reference values recommended by official agencies or in the literature; then, the medians were used as a cut-off point to dichotomize the MnH, MnTn, and PbB used as dependent variables to define the determinants associated with values above the median (considered as high level in this study). In the case of CdB, the cut-off points considered were the reference values for non-smoking adult Brazilian women (0.6 µg/L), and smokers (1 µg/L) were maintained [32,33,34].

The Chi-square test (χ^2^) was used to compare the frequencies of participants with high levels of biomarkers according to sociodemographic characteristics. Multivariate logistic regression (MLR) was used to investigate the relationship between the sociodemographic factors that showed association with the biomarkers of exposure to Mn (MnH and MnTn), Pb (PbB), and Cd (CdB) after dichotomization. Using Spearman’s correlation analysis, possible correlations were estimated between biomarkers (considered continuous variables).

Variables such as maternal age, schooling, gestational age, municipality of residence, and pre-gestational BMI were considered confounders and adjusted in the MLR based on the literature [35,36]. SPSS software version 23 for Windows was used for statistical analysis, and the significance level was *p* < 0.05.

## 3. Results

Of the 327 pregnant invited, 187 (57.2%) agreed to participate in the study (Figure 1). Due to dropouts, address changes, and spontaneous miscarriage, we collected information and biological samples from 163 (87.2%) pregnant women. However, depending on the variables collected in the questionnaires and biological samples, we had additional losses due to the unavailability or fear of donating biological samples (such as blood or hair).

### 3.1. Sociodemographic Characteristics of the Study Population

The sociodemographic data of the participants are presented in Table 1. The mean age of the participants was 27.0 (±6.1) years old, with an average of 18 weeks of pregnancy at the time of recruitment. Most pregnant women (94.5%) were self-declared as black or brown, and 43.6% were in their first pregnancy. More than two-thirds of the participants (68.2%) belong to a family with an income less than or equal to the minimum Brazilian wage (U$ 253), and 62.2% received assistance from the federal government. Less than half of the participants (46.6%) completed high school, with only 9.6% having completed higher education. More than half (52.4%) were classified as low SES (classes D and E “low” and “lowest”, respectively). Mean pre-gestational BMI was 25.1 kg/m^2^, with 5.6% and 15.1% classified as “underweight” and obese, respectively. One-third (32.5%) of the families burned garbage, while 13.9% renovated their homes during pregnancy. Almost 20% of pregnant women live with a smoker (husband or other relatives); few pregnant women were active smokers, 10.1% (*n* = 15) before pregnancy and only 2.2% (*n* = 2) during pregnancy. The participants presented the same social characteristics regardless of their municipality of residence, although some variables had significantly different distributions.

### 3.2. Descriptive of RtPb and Biomarkers

All the biomarkers and the RtPb (Table 1) showed a non-parametric distribution by the KS test and did not differ significantly according to the participant’s origin. In general, the medians (Q1–Q3) of RtPb, PbB, AsH, CdB, MnH and MnTn were, respectively, 13.0 (3.9–31.8) μg/m^2^/30 days; 0.9 (0.5–1.7) µg/dL; 0.02 (0.01–0.04) µg/g; 0.55 (0.1–0.9) µg/L; 0.2 (0.1–0.5) µg/g, and 0.6 (0.4–1.1) µg/g.

### 3.3. The Proportion of Pregnant Women with Levels above the Reference Values

Table 2 presents the proportions (95%; confidence interval) of pregnant women with biomarker exposure levels above the reference values. No pregnant woman had an AsH level above 1.0 µg/g [37], while few had a higher exposure to Mn measured in hair or toenails (2 to 7%) [38,39] and Pb in blood (5.1%) [40]. However, the proportions of participants with CdB higher than the reference values were moderate to high, being 22.2% (15.6–30.0) for smokers (1.0 μg/L), 46% (37.5–54.7) and 61.1% (52.4–69.3), respectively for non-smoking Brazilian (0.6 μg/L) and US (0.4 μg/L) women [32,33,34,41].

### 3.4. Associations between Exposure Biomarkers

Spearman’s correlation (Table 3) between exposure biomarkers showed a significant weak positive correlation between PbS and MnUp (rho = 0.240; *p* = 0.025).

### 3.5. Relationship between Exposure Biomarkers and Sociodemographic Variables

#### 3.5.1. AsH

The values of AsH were undetectable, being only 2.91% (*n* = 4) of the samples with AsH concentration above the detection limit (0.09 μg/L) of the method. Therefore, it was not possible to interpret the results.

#### 3.5.2. PbB

Only being primiparous was significantly associated (*p* = 0.018) with PbB concentrations. This relationship was confirmed by MLR (Table 4). Participants who had been pregnant in the past were 2.49 times more likely to have blood lead levels above the median.

#### 3.5.3. Mn (MnH and MnTn)

Mn biomarkers showed significant associations with SES, education, being an active or passive smoker, monthly income, receiving government subsidies, and domestic waste burning. However, after MLR (Table 5), only SES and passive cigarette smoking maintained a significant association in the case of MnH. MnTn levels were influenced by exposure to waste burning and SES. All models showed statistical significance.

#### 3.5.4. CdB

The MLR analysis summarized in Table 6 shows that that pregnant women whose families burned domestic waste and renovated their houses during the gestational period were, respectively, 3.47 and 9.21 more likely to have CdB levels above 0.6 μg/L. Being exposed to cigarette smoke increases the likelihood of having CdB levels above 1.0 μg/L by four times.

## 4. Discussion

In this exploratory study, we investigated the exposure to Pb, Cd, As, and Mn in biological samples (blood, hair, and toenails) of 163 pregnant women residing in two municipalities in the Recôncavo Baiano, Brazil. The exposure level to As was very low, with 98% of the participants having AsH below the method’s LOD. However, 4.3 and 5.1% of the participants had biomarkers above the recommended reference values for Mn and Pb. We found a worrying situation regarding exposure to Cd, given that 22.2 to 61.1% of the pregnant women had high levels depending on the reference value considered. This work demonstrates the importance and urgency of implementing, as in developed countries, biomonitoring strategies in the general population, especially in those of vulnerable groups such as pregnant women and children living in impoverished regions. The low SES, domestic waste burning, passive smoking, being multiparous, and having renovated the house were some of the determinant factors of the high levels of Mn, Pb, and Cd in this population.

### 4.1. Lead

Considering the environmental sample, the median RtPb (13.0 µg/m^2^/30 days) was much lower than the reference (431 µg/m^2^) suggested by the US Agency for Environmental Protection [43]. As our methodology considered the duration (30 days) of dust deposition, we estimated for the period of gestation the range of exposure of our study population, using the minimum and maximum range found (0.57 to 89.9 µg/m^2^/30 days). The results (5.13 to 802.71 µg/m^2^ for nine months) showed a probable situation of risk for these pregnant women.

Compared with another study carried out with school-age children in the municipality of Aratuípe, in a community producing Pb glazed ceramics (Maragogipinho district, Bahia, Brazil) [19], the median level observed was lower than that found in the exposed (169 μg/m^2^/30 days) or control (56.7 μg/m^2^/30 days) group. In the case of other published works that used a methodology similar to this study, our average RtPb was identical to that found in homes (17 μg/m^2^/30 days) in Germany [44] or daycare centers in Sydney, Australia (22 μg/m^2^/30 days) [45]. This finding also corroborates with some of the reported geometric means of RtPb (between 19 and 63 μg/m^2^/30 days) in elementary schools in Simões Filho, Bahia, Brazil [5].

PbB levels have been associated significantly with RtPb, as observed in several studies [19,45,46], showing evidence of the contribution of this matrix to human exposure. Although we found no correlation between RtPb and PbB, this possibility should be considered, as demonstrated by Ohtsu et al. [47].

The median (range) 0.95 (0.05–16.4) µg/dL of PbB was lower than the CDC recent reference value (3.5 µg/dL) [40,48]. Nonetheless, some participants presented a risky situation for themselves and their fetus, such as the participants with PbB levels above the reference value [40,42]. Also, considering only the limited number of pregnant women (*n* = 9) in the Maragogipinho district included in this research, the situation seems more worrying, since the median and mean PbB (3.07 and 1.9 μg/dL, respectively) were twofold higher than those found in the whole study population. It is essential to point out that these limits were not indicated as levels below which there are no toxic effects for humans, the pregnant woman, and the fetus [15,40,49].

In other studies, reporting exposure to low levels of Pb, the median found here was lower than the findings by Silver et al. [50], by La-Llave-León et al. [51], and Guy et al., [52] with a median (interquartile range) of 3.74 µg/dL (3.05–5.20). Despite the low level of PbB found by Guy et al. [52] in Benin, contrary to our investigation, almost half of the participants had PbB above the new CDC reference value. Few studies with pregnant women have reported levels lower than or similar to ours, such as Ohtsu et al. [47] in Japan, Wang et al. [53] and Perkins et al., ([54], in the USA). Despite this low level of PbB, Perkins et al. [54] reported the adverse effects of Pb on anthropometric outcomes in children born to pregnant women in their cohort. Some research has shown the negative CNS impacts of prenatal exposure to low levels of Pb in children. For example, Silver et al. [50] demonstrated an association of low levels of PbB with delayed maturation of the auditory and visual systems in neonates (with an average of two days of age), while Jedrychowski et al. [55] reported their association with low neurocognitive performance in a six-month-old baby.

Despite sufficient evidence in the literature on the relationship between socioeconomic factors and Pb exposure, only multiparity was associated with PbB levels above the median. The lack of relationship with other factors was also reported in other studies; for example, Ohtsu et al. [47] Taylor et al. [15] and Guy et al. [52] also found no association between PbB levels with factors such as exposure to cigarette smoking, SES, and the presence of peeling paint in the house. The low level of exposure, in addition to the absence of a specific exposure source, except for residents in Maragogipinho, maybe the reason, as reported by Taylor et al. [15].

Although only a significant influence of multiparity on PbB levels was detected, it is relevant to highlight the possible influences of some socioeconomic and environmental covariates. Passive smoking had a median PbB 1.28 times higher (1.02 vs. 0.80 µg/dL; *p* = 0.786) compared to those not exposed to cigarette smoke. Pregnant women from lower SES or education levels had higher medians when compared to another category of the same variable considered. In addition, considering only homes with peeling paint (*n* = 67), pregnant women who reported having a lot or moderate amount of peeling paint in their home had a median PbB (1.35 vs. 0.73 µg/dL; *p* = 0.24) 1.85 times higher than those with a little peeled amount. Taking into account the toxicity of Pb for the pregnant woman and the fetus and the existence of evidence in the literature, we suggest considering these factors in eventual awareness sessions for participants in our cohort and decision-makers in the respective municipalities.

### 4.2. Manganese

Few participants presented values of the Mn biomarkers above the reference values [38,39]. Ward et al. [39], suggested 4.14 µg/g as a cutoff point to distinguish welders exposed to Mn fumes from the control group of their study. Therefore, it is essential to consider that the reference values for these Mn biomarkers are also not related to the threshold compatible with the beneficial or deleterious effects of Mn. For the time being, there is little data on these matrices used as biomarkers either in the general population or in pregnant women [2,56,57]. Studies in populations with low levels are therefore essential to contribute to the correct definition of these reference values concerning possible effects; our cohort study has the final objective of estimating the impact of these exposures on children’s neurodevelopment at 12 months.

Considering studies carried out with pregnant women, Mora et al. [58], found a geometric mean of 1.8 µg/g in pregnant women’s hair in Costa Rica, while Rodrigues et al. [36], reported medians of 27.1 and 34.7 µg/g, respectively, in Pabna and Sirajdikhan (in Bangladesh). The difference is that these studies were conducted in areas close to sources of exposure (such as plantations using pesticides and drinking water from highly contaminated artesian wells), which could explain these levels above ours. Other studies carried out in children (15.2 µg/g) and in non-pregnant women (4.4 µg/g) in areas close to a ferromanganese alloy plant in Brazil also reported levels well above the ones reported here [3,4]. Even in the control group of their research, Menezes-Filho et al. [3] reported a median (1.76 µg/g) 8.8 times higher than those in this study. Regarding the evaluation of Mn in toenails, Signes-Pastor et al. [57] found in pregnant women a median (range) of 0.34 (0.17–0.72) µg/g, two times lower than that reported in this study. Our findings do not corroborate with those of Rodrigues et al. [36], probably due to the high levels of exposure.

Three factors associated with higher levels of Mn mainly stood out in the MLR: SES, waste burning, and passive exposure to cigarette smoke. However, most publications were carried out in populations with high exposure and mainly considered correlations or univariate analyzes [56], which makes comparisons difficult. Despite using hair and nails as matrices to estimate exposure to Mn, Viana et al. [4], found no correlation between Mn exposure and cigarette smoking or waste burning. Although Tasker et al. [10] used blood as a matrix in their investigation, the authors reported a relationship between second hand cigarette smoking and blood Mn levels (MnB) in the second trimester. Still, their finding was contrary to ours (lower levels of MnB in smokers). Contrary to our findings, Viana et al. [4] reported significantly higher median levels of MnH in individuals with less education.

Co-exposure to Pb and Mn: Of the metals evaluated, only Pb and Mn (PbB and MnTn) showed a correlation in their exposure dynamics. As the matrices (toenails and blood) do not suggest exposure simultaneously (7 to 12 months and 1-month window for toenails and blood [4,57], respectively), we suggest that this relationship is spurious and not related to any biological mechanism absorption of these metals.

### 4.3. Cadmium

CdB level is a biomarker of recent exposure to Cd, in addition to being a good proxy for chronic exposure to low levels; that is, even in a population not occupationally exposed, it is possible to find worrying levels [59,60]. The CdB values found in our study population point to a situation contrary to our expectations.

Mean and median values in this study exceeded CdB levels in workers and populations living close to areas with a source of exposure. Ferron et al. [6] reported in recycling sorting workers from São Paulo an arithmetic mean (0.47 μg/L) and median (0.44 μg/L) lower than our values (respectively, 0.81 and 0.55 µg/L). The control group of this study presented an average value 6.7 times lower than ours. Also worrying was the maximum value (7.61 μg/L), which is well above what was found by Ferron et al. [6] (2020) and Naka et al., [61]. In non-pregnant women residing in the industrial zone of the state of Amazonas [61], the authors also found a similar average (0.46 μg/L) while the control group presented a value (0.22 μg/L) 2.5 times lower.

Some studies conducted with general populations outside Brazil reported similar situations to ours. For example, Garner and Levallois [35] in Canada reported a geometric mean of 0.43 μg/L in non-pregnant women, similar to Sakellari et al. [62] in Greece and Johntson et al. [60] from the USA. According to the authors [35,60], cigarette smoking may be the leading cause of the higher level of Cd. Indeed, as expected, passive exposure to cigarette smoke was significantly associated with a fourfold higher chance of having levels above 1 μg/L. Active smoking before pregnancy did not show a significant relationship with high levels of CdB, given the small number of observations. Passive cigarette smoking exposure is a relevant source of contamination of a multitude of compounds, which raises the importance of the shared responsibility of all residents in a home for the health of the pregnant woman and her fetus. Besides the pregnant woman stopping smoking, the other house residents would have to do the same or avoiding smoking indoors. Active or passive smoking status was self-reported by participants; the fact of not having evaluated cotinine, a biomarker of exposure to nicotine [60] may be a limitation of our study. Nevertheless, pregnancy is often a reason that leads many women to stop smoking or reduce their habit.

House renovation (or painting) was another determining factor that increased the likelihood of having CdB levels above 1 or 0.6 μg/L. We suggest that the type of paint used during the renovation could explain this finding. In fact, during our visits to the participants’ homes, we noticed the reforms or changes in the residences to adjust for the arrival of the baby. Some works have shown the possibility of finding high levels of cadmium and other metals in paint pigments or building and renovation materials [63,64]. Other possible sources of exposure to Cd could be living close to or inside workshops or production units of palm oil, soap (some participants), or even a diet rich in shellfish or animal viscera. We have visited pregnant women whose homes were located in these environments. Regarding diet, data on food frequency is being processed.

None of the sociodemographic variables (SES, age, education, income, municipality) and the fact of not being primiparous were not associated with an increase in CdB above the reference values. Contrary to what was expected with alcohol consumption [35,60,65], no association was detected, probably because pregnant women stopped drinking when they were aware of their pregnancy.

### 4.4. Strengths and Limitations

As a cross-sectional study design, this work carries some limitations of this type of design, such as recall and nonresponse biases. For example, some biological samples were collected from only some participants; some refused to collect certain biological samples (blood or hair), while others were unavailable during the second semester. That may explain why there was no association in some statistical analyses. Despite the study design, our research was able to evaluate the current and chronic exposure to Mn as we evaluated its level in hair and toenails. Also, this work is one of the few carried out in developing countries with vulnerable populations; it adds evidence to sustain the urgency of implementing biomonitoring and vigilance of PTM exposure in Brazil.

## 5. Conclusions

Despite the low levels of exposure in general, this study showed worrying Cd exposure levels of most participants and reported the main risk factors of exposure to the PTM studied. Considering the evidence of the deleterious effects of low PTM levels on fetal development and the health of pregnant women, it is useful to investigate further. As a scientific community, there is a need to adopt a stronger position for implementing a biomonitoring policy in vulnerable populations.

## Figures and Tables

**Figure 1 ijerph-20-02949-f001:**
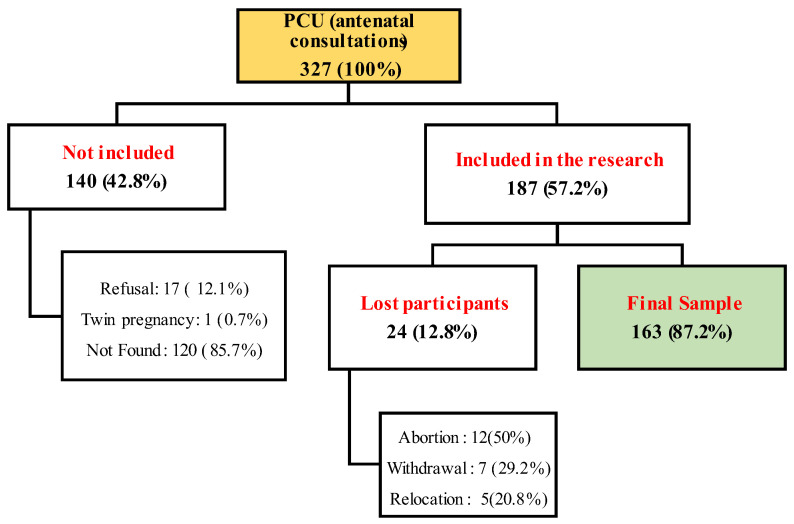
Selection flowchart of the population study.

**Table 1 ijerph-20-02949-t001:** Sociodemographic and descriptive characteristics of the evaluated biomarkers of the study population.

Variables	Categories: *n* (%)			
Ethnicity	Black/Brown	154 (94.5)	White	9 (5.5)
Marital status	Married/Stable Union	89 (54.6)	Single/divorced	74 (45.4)
SES	D/E	77 (52.4)	C/B	70 (47.6)
Family income	Up to 1 salary	90 (68.2)	Above 1 salary	42 (31.8)
Government assistance	Yes	84 (62.2)	No	51 (37.8)
Education	Up to elementary school	87 (53.4)	High school and higher	76 (46.6)
Occupation	Housewife	51 (31.3)	Autonomous/other	112 (68.7)
House renovated	Yes	20 (13.9)	No	124 (86.1)
Passive smoker	Yes	32 (19.5)	No	112 (68.3)
Waste burning	Yes	53 (32.5)	No	110 (67.5)
First parity	Yes	71 (43.6)	No	92 (56.4)
		N	Mean ± SD	Median (Q1–Q3)
Age (years)		163	27.0 ± 6.1	26.8 (22.1–31.9)
Gestational age at inclusion (Weeks)		163	18.1 ± 5.3	18 (14.00–22.00)
Pre-gestational BMI (kg/m^2^)		126	25.1 ± 4.4	24.8 (22.1–28.03)
RtPb (μg/m^2^/30 days)		52	20.8 ± 22.2	13.0 (3.9–31.8)
PbB (μg/dL)		117	1.42 ± 2.23	0.9 (0.5–1.7)
MnH (μg/g)		131	0.4 ± 0.5	0.2 (0.1–0.5)
MnTn (μg/g)		105	0.9 ± 0.9	0.6 (0.4–1.1)
AsH (μg/g)		137	0.03 ± 0.02	0.02 (0.01–0.04)
CdB (μg/L)		126	0.8 ± 1.0	0.55 (0.1–0.9)

SD: Standard Deviation.

**Table 2 ijerph-20-02949-t002:** Proportion of pregnant women with PTM levels above the references.

	*n*	Freq. (prop. %)	(95% CI)	References
Cd				
CdB ≥ 0.4 μg/L	126	77 (61.1)	52.4–69.3	[41]
CdB ≥ 0.6 μg/L	126	58 (46.0)	37.5–54.7	[32,33]
CdB ≥ 1 μg/L	126	28 (22.2)	15.6–30.0	[34]
Mn				
MnH ≥ 1.2 μg/g	131	7 (4.3)	2.3–10.1	[38]
MnTn ≥ 4.14 μg/g	105	2 (1.9)	0.3–5.8	[39]
As				
AsH ≥ 1 μg/g	137	0 (0)	0–1.4	[37]
Pb				
PbB ≥ 3.5 μg/dL	117	6 (5.1)	2.1–10.1	[40]
PbB ≥ 2.0 μg/dL	117	23 (19.7)	13.2–27.5	[42]

Prop.: proportion.

**Table 3 ijerph-20-02949-t003:** Spearman correlation matrix between environmental markers and exposure biomarkers.

	RtPb	MnH	MnTn	PbB	CdB
RtPb					
Rho	1.000	0.158	0.045	−0.216	0.012
P		0.268	0.776	0.159	0.938
N		51	42	44	46
MnH					
Rho		1.000	0.201 *	−0.063	0.010
P			0.044	0.486	0.911
N			101	126	116
MnTn					
Rho			1.000	0.240 *	0.121
P				0.025	0.245
N				87	94
PbB					
Rho				1.000	0.133
P					0.154
N					117
CdB					
Rho					1.000
P					
N					

* Correlation significant at 0.05 (bilateral).

**Table 4 ijerph-20-02949-t004:** Determinants of PbB levels above the median after χ^2^ and MLR test.

	PbS < 0.9 μg/dL	PbS ≥ 0.9 μg/dL	OR (95% CI)	*p*-Value
	*n* (%)	*n* (%)		
Education				
≤High school	26 (44.8)	32 (55.2)	Ref.	
≥Elementary school	33 (55.9)	26 (44.1)	0.59 (0.24–1.45)	0.245
Waste burning				
No	46 (53.5)	40 (46.5)	Ref.	
Yes	13 (41.9)	18 (58.1)	2.06 (0.76–5.58)	0.153
First parity				
Yes	31 (63.3)	18 (36.7)	Ref.	
No	28 (41.2)	40 (58.8)	2.49 (1.02–6.07)	0.045

Adjusted for BMI, municipality, age, gestational age, education.

**Table 5 ijerph-20-02949-t005:** Determinants of Mn biomarker levels above the median after χ^2^ test and MLR analysis.

	MnUp < 0.6 μg/g	MnUp ≥ 0.6 μg/g	OR (95% CI)	*p*-Value
	*n* (%)	*n* (%)		
Active smoking				
No	51 (54.8)	42 (45.2)	Ref.	
Yes	2 (18.2)	9 (81.8)	2.62 (0.40–17.32)	0.318
SES				
C/B/A	35 (67.3)	17 (32.7)	Ref.	
D/E	16 (32.7)	33 (67.3)	4.23 (1.70–10.53)	0.002
Waste burning				
No	45 (58.4)	32 (41.6)	Ref.	
Yes	7 (26.9)	19 (73.1)	5.10 (1.43–17.69)	0.010
	MnC < 0.2 μg/g	MnC ≥ 0.2 μg/g	OR (95% CI)	
	*n* (%)	*n* (%)		
SES				
C/B/A	42 (67.7)	20 (32.3)	Ref.	
D/E	23 (35.9)	41 (64.1)	3.95 (1.79–8.73)	0.001
Passive smoking				
No	56 (58.3)	40 (41.7)	Ref.	
Yes	10 (34.5)	19 (65.5)	2.62 (1.004–6.835)	0.049

Adjusted for BMI, municipality, age, gestational age, education; SES: socioeconomic status.

**Table 6 ijerph-20-02949-t006:** Determinants of CdB level above reference values based on χ^2^ and MLR.

	CdS< 0.6 μg/L	CdS ≥ 0.6 μg/L	OR (95% CI)	*p*-Value
	*n* (%)	*n* (%)		
Passive smoking				
No	54 (58.7)	38 (41.3)	Ref.	
Yes	12 (40.0)	18 (60.0)	2.95 (0.98–8.85)	0.054
SES				
C/B/A	35 (60.3)	23 (39.7)	Ref.	
D/E	31 (49.2)	32 (50.8)	0.69 (0.23–2.05)	0.501
Waste burning				
No	54 (59.3)	37 (40.7)	Ref.	
Yes	13 (39.4)	20 (60.6)	3.47 (1.09–11.05)	0.035
House renovation				
No	59 (57.3)	44 (42.7)	Ref.	
Yes	5 (29.4)	12 (70.6)	9.21 (1.90–44.57)	0.006
	CdS < 1.0 μg/L	CdS ≥ 1.0 μg/L		
	*n* (%)	*n* (%)		
Passive smoking				
No	76 (82.6)	16 (17.4)	Ref.	
Yes	19 (63.3)	11 (36.7)	4.01 (1.24–13.038)	0.021
House renovation				
No	83 (80.6)	20 (19.4)	Ref.	
Yes	9 (52.9)	8 (47.1)	7.02 (1.78–27.37)	0.005

Adjusted for BMI, municipality, age, gestational age, education.

## Data Availability

Not applicable.

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
