# Peer review of "Determinants of Exposure to Potentially Toxic Metals in Pregnant Women of the DSAN-12M Cohort in the Recôncavo Baiano, Brazil"

_ijerph, 2023, doi:10.3390/ijerph20042949_

Round 1

Reviewer 1 Report

Homègnon A. Ferréol Bah and colleagues reports the study of toxic metals on pregnant women and child health in DSAN-12M cohort in the Recôncavo Baiano, Brazi. Though the toxic effect of harmful metals is known but the information in present study is important and relevant, considering the size of the effects. In general, the work is potentially helpful for clinical as well as basic science understanding. The manuscript is well written and scientifically sound. Manuscript is acceptable with the following minor comments:

1. Figure 1 is not needed, can be removed.

2. Please correct the Figure number from flow chart, it is Fig 2.

3. Please correct the flow chart, coma sign should be removed from all % presentation, e.g. instead of 42,2%, it should be 42.2%, I believe.

4. Methodology, results and discussion sections are too descriptive, somewhat limiting the readability of the article, this could be shortened. Authors should leave essential to describe the study, if necessary could be inserted some summary tables of the results.

Author Response

Response to Reviewer Comments

Homègnon A. Ferréol Bah and colleagues reports the study of toxic metals on pregnant women and child health in DSAN-12M cohort in the Recôncavo Baiano, Brazil. Though the toxic effect of harmful metals is known but the information in present study is important and relevant, considering the size of the effects.

In general, the work is potentially helpful for clinical as well as basic science understanding. The manuscript is well written and scientifically sound. Manuscript is acceptable with the following minor comments:

Point 1: Figure 1 is not needed, can be removed.

Response 1: Thank you for the suggestion; we have considered your recommendation. The figure 1 was removed.

Point 2: The Please correct the Figure number from flow chart, it is Fig 2.

Response 2: Thank you for the suggestion; we have considered your recommendation. Now you can read: Fig 1 (as the former figure 1 was removed; Fig 2 is now Fig 1)

Point 3: Please correct the flow chart, coma sign should be removed from all % presentation, e.g. instead of 42,2%, it should be 42.2%, I believe.

Response 3: Thank you for observing this mistake. We made the correction accordingly.

 Point 4: Methodology, results and discussion sections are too descriptive, somewhat limiting the readability of the article, this could be shortened. Authors should leave essential to describe the study, if necessary could be inserted some summary tables of the results.

Response 4: Thank you for the suggestion; we have considered your recommendation and revising this article. We marked in yellow the parts corrected in the document. 

Reviewer 2 Report

This paper is very popular for health and environment that many researchers are interested. I have read two or three times very carefully, may be the authors can clarify the method(s)  a little more.

I try to understand  but for the first readers, maybe is difficult.

Materials and analyses required can be clarifed..

 Another point is the conclusion. You have the results detailed. It is better to summarized  this part in conclusion.

Author Response

Response to Reviewer Comments

This paper is very popular for health and environment that many researchers are interested. I have read two or three times very carefully, may be the authors can clarify the method(s) a little more.

I try to understand but for the first readers, maybe is difficult.

 Point 1: Materials and analyses required can be clarifed

 Response 1: Thank you for the suggestion; we have considered your recommendation. We add a short introduction to explain briefly the section “2.2. Assessment of exposure to PTM”. Also, we reviewed the whole section “2. Material and methods” to make more easier to the reader. We marked in yellow the part corrected.  

Point 2: Another point is the conclusion. You have the results detailed. It is better to summarized this part in conclusion.

Response 2: Thank you for the suggestion; we have considered your recommendation and reviewed the conclusion. The part corrected is marked in yellow.

Reviewer 3 Report

The abstract is adequately written mentioning all the aspects of the work. It is very simplified focused on main results of the study.

The introduction is nice. this section requires to cover some aspects such as problem definition and statement, literature review conducted regional or global and structure of the paper.

The methods are presented adequately but still lacks with statistical description of methods implemented.

The results and discussion section needs improvement in some sections particularly discussion on the impact of differnt metals should be elaborately dsicussed.

the conclusions should be redefined again following the results obtained.

overall, i have no objection in accepting the paper following the implementation or amendments of the above provided suggestions before the submission of revised paper.

Author Response

Response to Reviewer Comments

Point 1: The abstract is adequately written mentioning all the aspects of the work. It is very simplified focused on main results of the study.

Response 1: Thank you for the comments.

Point 2: The introduction is nice. this section requires to cover some aspects such as problem definition and statement, literature review conducted regional or global and structure of the paper.

Response 2: Thank you for the suggestion; we have considered it. We edited the introduction upon your recommendation along with the revision of the whole manuscript. The part corrected is marked in yellow.

Point 3: The methods are presented adequately but still lacks with statistical description of methods implemented.

Response 3: Thank you for the observation; we have considered your comment and edited this section. The part corrected is marked in yellow.

Point 4: The results and discussion section need improvement in some sections particularly discussion on the impact of different metals should be elaborately discussed.

Response 4: Thank you for the comments. We have provided some corrections in those sections considering your suggestions. However, in the discussion we did not discuss in detail the impact of different metals as you suggested to avoid making this section too lengthy. Issues raised by other reviewers of this manuscripts, the focus of paper and for the sake of conciseness limited the space for more detailed discussion.

Point 5: The conclusions should be redefined again following the results obtained.

Response 5: Thank you for the suggestion; we have considered your recommendation and reviewed the conclusion. The text edited is marked in yellow.
